# MULTITASK SOFT OPTION LEARNING

## ABSTRACT

We present Multitask Soft Option Learning (MSOL), a hierarchical multitask framework based on Planning as Inference. MSOL extends the concept of options, using separate variational posteriors for each task, regularized by a shared prior. This allows fine-tuning of options for new tasks without forgetting their learned policies, leading to faster training without reducing the expressiveness of the hierarchical policy. MSOL avoids several instabilities during training in a multitask setting and provides a natural way to learn both intra-option policies and their terminations. We demonstrate empirically that MSOL significantly outperforms both hierarchical and flat transfer-learning baselines in challenging multi-task environments.

## 1 INTRODUCTION

A key challenge in Reinforcement Learning (RL) is to scale current approaches to higher complexity tasks without requiring a prohibitive number of environmental interactions. However, for many tasks, it is possible to construct or learn efficient exploration priors that allow to focus on more relevant parts of the state-action space, reducing the number of required interactions. These include, for example, reward shaping (Ng et al., 1999; Konidaris & Barto, 2006), curriculum learning (Bengio et al., 2009), some meta-learning algorithms (Wang et al., 2016; Duan et al., 2016; Gupta et al., 2018; Houthooft et al., 2018; Xu et al., 2018), and transfer learning (Caruana, 1997; Taylor & Stone, 2011; Bengio, 2012; Parisotto et al., 2015; Rusu et al., 2015; Teh et al., 2017).

One promising way to capture prior knowledge is to decompose policies into a hierarchy of sub-policies (or skills) that can be reused and combined in novel ways to solve new tasks (Dayan & Hinton, 1993; Thrun & Schwartz, 1995; Parr & Russell, 1998; Sutton et al., 1999; Barto & Mahadevan, 2003). The idea of Hierarchical RL (HRL) is also supported by findings that humans appear to employ a hierarchical mental structure when solving tasks (Botvinick et al., 2009; Collins & Frank, 2016). In such a hierarchical RL policy, lower-level, *temporally extended* skills yield directed behavior over multiple time steps. This has two advantages: i) it allows efficient exploration, as the target states of skills can be reached without having to explore much of the state space in between, and ii) directed behavior also reduces the variance of the future reward, which accelerates convergence of estimates thereof. On the other hand, while a hierarchical approach can therefore significantly speed up exploration and training, it can also severely limit the expressiveness of the final policy and lead to suboptimal performance when the temporally extended skills are not able to express the required policy for the task at hand (Mankowitz et al., 2014).

Many methods exist for constructing and/or learning skills for particular tasks (Dayan & Hinton, 1993; Sutton et al., 1999; McGovern & Barto, 2001; Menache et al., 2002; Şimşek & Barto, 2009; Gregor et al., 2016; Kulkarni et al., 2016; Bacon et al., 2017; Nachum et al., 2018a). Training on multiple tasks simultaneously is one promising approach to learn skills that are both relevant and generalise across tasks (Thrun & Schwartz, 1995; Pickett & Barto, 2002; Fox et al., 2016; Andreas et al., 2017; Frans et al., 2018). Ideally, the entire hierarchy can be trained end-to-end on the obtained return, obviating the need to specify proxy rewards for skills (Frans et al., 2018).

However, learning hierarchical policies end-to-end in a multitask setting poses two major challenges: i) because skills optimize environmental rewards directly, correctly updating them relies on already (nearly) converged master policies that use them similarly across all tasks, requiring complex training schedules (Frans et al., 2018), and ii) the end-to-end optimization is prone to local minima in

which multiple skills have learned similar behavior. This second points is explained in more detail in Appendix A.

In this paper, we propose Multitask Soft Option Learning (MSOL), a novel approach to learning hierarchical policies in a multi-task setting that extends Options (Sutton et al., 1999), a common definition for skills, and casts the concept into the Planning as Inference (PAI) framework (see, e.g., Levine, 2018, for a review). MSOL brings multiple advantages: i) it stabilizes end-to-end multitask training, removing the need for complex training schedules like in Frans et al. (2018), ii) it gives rise to coordination between master policies, avoiding local minima of the type described in Appendix A, iii) it allows fine-tuning of options, i.e. adapting them to new tasks at test-time without the risk of unlearning previously acquired useful behavior, thereby avoiding suboptimal performance due to restricted expressiveness, iv) and lastly, we show how the *soft option* framework gives rise to a natural solution to the challenging task of learning option-termination policies.

MSOL differentiates between a prior policy for each option, shared across all tasks, and a flexible task-specific posterior policy. The option prior can be fixed once it is fully trained, preventing un-learning of useful behavior even when the posteriors are updated. On new tasks, the option posteriors are initialized to the priors and regularized towards them, but are still adaptable to the specific task. This allows the same accelerated training as with 'hard' options, but can solve more tasks due to the adjustable posteriors. Furthermore, during option learning, we can train prior and posterior policies simultaneously (Teh et al., 2017), all without the need for complex training schedules (Frans et al., 2018): training is stabilized because only the priors are shared across tasks.

Our experiments demonstrate that MSOL outperforms previous hierarchical and transfer learning algorithms during transfer tasks in a multitask setting. MSOL only modifies the regularized reward and loss function, but does not require any specialized architecture. In particular, it also does not require artificial restrictions on the expressiveness of either the higher-level or intra-option policies.

## 2 PRELIMINARIES

An agent's task is formalized as a MDP $(\mathcal{S}, \mathcal{A}, \rho, P, r)$, consisting of the state space $\mathcal{S}$, the action space $\mathcal{A}$, the initial state distribution $\rho$, the transition probability $P(s_{t+1}|s_t, a_t)$ of reaching state $s_{t+1}$ by executing action $a_t$ in state $s_t$, and the reward $r(s_t, a_t)$ an agent receives for this transition.

### 2.1 PLANNING AS INFERENCE

Planning as inference (PAI) (Todorov, 2008; Toussaint, 2009; Kappen et al., 2012) frames RL as a probabilistic-inference problem (Levine, 2018). The agent learns a distribution $q_\phi(a|s)$ over actions $a$ given states $s$, i.e., a policy, parameterized by $\phi$, which induces a distribution over trajectories $\tau$ of length $T$, i.e., $\tau = (s_1, a_1, s_2, \ldots, a_T, s_{T+1})$:

$$q_\phi(\tau) \;\;=\;\; \rho(s_1)\prod_{t=1}^{T} q_\phi(a_t|s_t)\, P(s_{t+1}|s_t, a_t)\,. \tag{1}$$

This can be seen as a structured variational approximation of the optimal trajectory distribution. Note that the true initial state probability $\rho(s_1)$ and transition probability $P(s_{t+1}|s_t, a_t)$ are used in the variational posterior, as we can only control the policy, not the environment.

An advantage of this formulation is that we can incorporate information both from prior knowledge, in the form of a prior policy distribution, and the task at hand through a likelihood function that is defined in terms of the achieved reward. The prior policy $p(a_t|s_t)$ can be specified by hand or, as in our case, learned (see Section 3). To incorporate the reward, we introduce a binary *optimality variable* $\mathcal{O}_t$ (Levine, 2018), whose likelihood is highest along the optimal trajectory that maximizes return: $p(\mathcal{O}_t = 1|s_t, a_t) = \exp\left(r(s_t, a_t)/\beta\right)$. The constraint $r \in (-\infty, 0]$ can be relaxed without changing the inference procedure (Levine, 2018). For brevity, we denote $\mathcal{O}_t = 1$ as $\mathcal{O}_t \equiv (\mathcal{O}_t = 1)$. If a given prior policy $p(a_t|s_t)$ explores the state-action space sufficiently, then $p(\tau, \mathcal{O}_{1:T})$ is the distribution of desirable trajectories. PAI aims to find a policy such that the variational posterior in (1) approximates this distribution by minimizing the Kullback-Leibler (KL) divergence

$$\mathcal{L}(\phi) = \mathbb{D}_{\text{KL}}(q_\phi(\tau) \,\|\, p(\tau, \mathcal{O}_{1:T}))\,, \text{ with } p(\tau, \mathcal{O}_{1:T}) = \rho(s_1)\prod_{t=1}^{T} p(a_t|s_t)P(s_{t+1}|s_t, a_t)p(\mathcal{O}_t|s_t, a_t). \tag{2}$$

## 2.2 MULTI-TASK LEARNING

In a multi-task setting, we have a set of different tasks $i \in \mathcal{T}$, drawn from a task distribution with probability $\xi(i)$. All tasks share state space $\mathcal{S}$ and action space $\mathcal{A}$, but each task has its own initial-state distribution $\rho_i$, transition probability $P_i(s_{t+1}|s_t, a_t)$, and reward function $r_i$. Our goal is to learn $n$ tasks concurrently, distilling common information that can be leveraged to learn faster on new tasks from $\mathcal{T}$. In this setting, the prior policy $p_\theta(a_t|s_t)$ can be learned jointly with the task-specific posterior policies $q_{\phi_i}(a_t|s_t)$ (Teh et al., 2017). To do so, we simply extend (2) to

$$\mathcal{L}(\{\phi_i\}, \theta) \quad = \quad \mathbb{E}_{i \sim \xi}\left[\, \mathbb{D}_{\text{KL}}(q_{\phi_i}(\tau) \,\|\, p_\theta(\tau, \mathcal{O}_{1:T}))\,\right] \quad = \quad -\frac{1}{\beta}\, \mathbb{E}_{i \sim \xi, \tau \sim q}\left[\sum_{t=1}^{T} R_{i,t}^{\text{reg}}\right], \quad (3)$$

where $R_{i,t}^{\text{reg}} := r_i(s_t, a_t) - \beta \ln \frac{q_{\phi_i}(a_t|s_t)}{p_\theta(a_t|s_t)}$ is the regularised reward. Minimizing the loss in (3) is equivalent to maximizing the regularized reward $R_{i,t}^{\text{reg}}$. Moreover, minimizing the term $\mathbb{E}_{\tau \sim q}\left[\ln \frac{q_{\phi_i}(a_t|s_t)}{p_\theta(a_t|s_t)}\right]$ implicitly minimizes the expected KL-divergence $\mathbb{E}_{s_t \sim q}\left[\mathbb{D}_{\text{KL}}[q_{\phi_i}(\cdot|s_t)\|p_\theta(\cdot|s_t)]\right]$. In practise (see Appendix C.1) we will also make use of a discount factor $\gamma \in [0, 1]$. For details on how $\gamma$ arises in the PAI framework we refer to Levine (2018).

## 2.3 OPTIONS

Options (Sutton et al., 1999) are skills that generalize primitive actions and consist of three components: i) an intra-option policy $p(a_t|s_t, z_t)$ selecting primitive actions according to the currently active option $z_t$, ii) a probability $p(b_t|s_t, z_{t-1})$ of terminating the *previously* active option $z_{t-1}$, and iii) an initiation set $\mathcal{I} \subseteq \mathcal{S}$, which we simply assume to be $\mathcal{S}$. Note that by construction, the higher-level (or master-) policy $p(z_t|z_{t-1}, s_t, b_t)$ can only select a new option $z_t$ if the previous option $z_{t-1}$ has terminated.

## 3 METHOD

We aim to learn a reusable set of options that allow for faster training on new tasks from a given distribution. We learn both intra-option and termination policies, while preventing multiple options from learning the same behavior.

To differentiate ourselves from classical 'hard' options, which, once learned, do not change during new tasks, we call our novel approach *soft-options* (this is further discussed in Appendix B). Each soft-option consists of an option *prior*, which is shared across all tasks, and a task-specific option *posterior*. The priors of both the intra-option policy and the termination policy capture how an option typically behaves and remain fixed once they are fully learned. At the beginning of training on a new task, they are used to initialize the task-specific posterior distribution. During training, the posterior is then regularized against the prior to prevent inadvertent unlearning. However, if maximizing the reward on certain tasks is not achievable with the prior policy, the posterior is free to deviate from it. We can thus speed up training using options, while remaining flexible enough to solve any task.

Additionally, this soft option framework also allows for learning good *priors* in a multitask setting while avoiding local minima in which several options learn the same behavior. See Figure 1 for an overview over the hierarchical prior-posterior architecture that we explain further below.

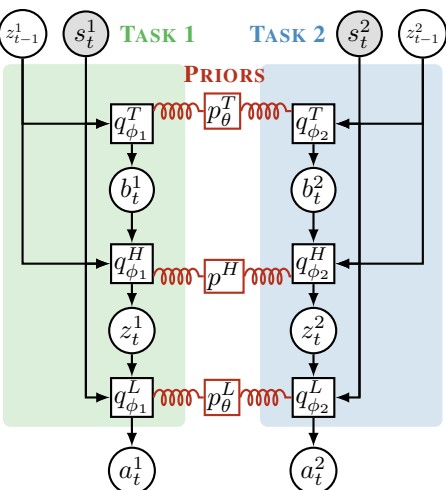

Figure 1: Two hierarchical posterior policies (left and right) with common priors (middle). For each task $i$, the policy conditions on the current state $s_t^i$ and the last selected option $z_{t-1}^i$. It samples, in order, whether to terminate the last option ($b_t^i$), which option to execute next ($z_t^i$) and what primitive action ($a_t^i$) to execute in the environment.

### 3.1 HIERARCHICAL POSTERIOR POLICIES

To express options in the PAI framework, we introduce two additional variables at each time step $t$: *option selections* $z_t$, representing the currently selected option, and decisions $b_t$ to *terminate* them and allow the higher-level (master) policy to choose a new option. The agent's behavior depends on the currently selected option $z_t$, by drawing actions $a_t$ from the *intra-option posterior policy* $q_{\phi_i}^L(a_t|s_t,z_t)$. The selection $z_t$ itself is drawn from a *master policy* $q_{\phi_i}^H(z_t|s_t,z_{t-1},b_t) = (1 - b_t)\,\delta(z_t - z_{t-1}) + b_t\, q_{\phi_i}^H(z_t|s_t)$, which conditions on $b_t \in \{0,1\}$, drawn by the *termination posterior policy* $q_{\phi_i}^T(b_t|s_t,z_{t-1})$. The master policy either continues with the previous $z_{t-1}$ or draws a new option, where we set $b_1 = 1$ at the beginning of each episode. We slightly abuse notation by referring by $\delta(z_t - z_{t-1})$ to the Kronecker delta $\delta_{z_t,z_{t-1}}$ for discrete and the Dirac delta distribution for continuous $z_t$. The joint posterior policy is

$$q_{\phi_i}(a_t,z_t,b_t|s_t,z_{t-1}) \quad = \quad q_{\phi_i}^T(b_t|s_t,z_{t-1})\, q_{\phi_i}^H(z_t|s_t,z_{t-1},b_t)\, q_{\phi_i}^L(a_t|s_t,z_t). \tag{4}$$

While $z_t$ can be a continuous variable, we consider only $z_t \in \{1\ldots m\}$, where $m$ is the number of available options. The induced distribution $q_{\phi_i}(\tau)$ over trajectories of task $i$, $\tau = (s_1,b_1,z_1,a_1,s_2,\ldots,s_T,b_T,z_T,a_T,s_{T+1})$, is then

$$q_{\phi_i}(\tau) \quad = \quad \rho_i(s_1)\prod_{t=1}^{T} q_{\phi_i}(a_t,z_t,b_t|s_t,z_{t-1})P_i(s_{t+1}|s_t,a_t). \tag{5}$$

### 3.2 HIERARCHICAL PRIOR POLICY

Our framework transfers knowledge between tasks by a shared prior $p_\theta(a_t,z_t,b_t|s_t,z_{t-1})$ over all joint policies (4): $p_\theta(a_t,z_t,b_t|s_t,z_{t-1}) = p_\theta^T(b_t|s_t,z_{t-1})\, p^H(z_t|z_{t-1},b_t)\, p_\theta^L(a_t|s_t,z_t)$. By choosing $p_\theta^T$, $p^H$, and $p_\theta^L$ correctly, we can learn useful temporally extended options. The parameterized priors $p_\theta^T(b_t|s_t,z_{t-1})$ and $p_\theta^L(a_t|s_t,z_t)$ are structurally equivalent to the posterior policies $q_{\phi_i}^T$ and $q_{\phi_i}^L$ so that they can be used as initialization for the latter. Optimizing the regularized return (see next section) w.r.t. $\theta$ distills the common behavior into the prior policy and softly enforces similarity across posterior distributions of each option amongst all tasks $i$.

The prior $p^H(z_t|z_{t-1},b_t) = (1-b_t)\,\delta(z_t - z_{t-1}) + b_t\frac{1}{m}$ selects the previous option $z_{t-1}$ if $b_t = 0$, and otherwise draws options uniformly to ensure exploration. Because the posterior master policy is different on each task, there is no need to distill common behavior into a joint prior.

### 3.3 OBJECTIVE

We extend the multitask objective in (3) by substituting $p_\theta(\tau,\mathcal{O}_{1:T})$ and $p_{\phi_i}(\tau)$ with those induced by our hierarchical posterior policy in (4) and the corresponding prior. The resulting objective has the same form but with a new regularized reward that is maximized:

$$R_{i,t}^{\text{reg}} \quad := \quad r_i(s_t,a_t) - \underbrace{\beta\ln\frac{q_{\phi_i}^H(z_t|s_t,z_{t-1},b_t)}{p^H(z_t|z_{t-1},b_t)}}_{①} - \underbrace{\beta\ln\frac{q_{\phi_i}^L(a_t|s_t,z_t)}{p_\theta^L(a_t|s_t,z_t)}}_{②} - \underbrace{\beta\ln\frac{q_{\phi_i}^T(b_t|s_t,z_{t-1})}{p_\theta^T(b_t|s_t,z_{t-1})}}_{③}. \tag{6}$$

As we maximize $\mathbb{E}_q[R_{i,t}^{\text{reg}}]$, this corresponds to maximizing the expectation over $r_i(s_t,a_t) - \beta\big[\mathbb{D}_{\text{KL}}(q_{\phi_i}^H\|p^H) + \mathbb{D}_{\text{KL}}(q_{\phi_i}^L\|p_\theta^L) + \mathbb{D}_{\text{KL}}(q_{\phi_i}^T\|p_\theta^T)\big]$, along the on-policy trajectories drawn from $q_{\phi_i}(\tau)$. Term ① of the regularization encourages exploration in the space of options. It can *also* be seen as a form of deliberation cost (Harb et al., 2017) as it is only nonzero whenever we terminate an option and the master policy needs to select another to execute.

Term ② softly enforces similarity between option posteriors across tasks and updates the prior towards the 'average' posterior. It also encourages the master policy to pick the most *specialized* option whose posteriors across all tasks are most similar. In other words, the master policy $q_{\phi_i}^H$ is encouraged to pick option $z_t$ which maximizes $r_i$, but minimizes term ② by picking the option $z_t$ for which prior $p_\theta^L$ and posterior $q_{\phi_i}^L$ are the most similar. Because the prior is the average posterior, this rewards the master policy to pick the most specialized option (that still achieves high reward). As discussed in more detail in Appendix A, this allows us to escape the local optimization minima that hard options face in multitask learning, while still having fully specialized options after training.

Lastly, we can use ③ to also encourage temporal abstraction of options. To do so, *during option learning*, we fix the termination prior $p^T$ to a Bernoulli distribution $p^T(b) = (1-\alpha)^b \alpha^{1-b}$. Choosing a large $\alpha$ encourages prolonged execution of one option, but allows switching whenever necessary. This is also similar to deliberation costs (Harb et al., 2017) but with a more flexible cost model.

Additionally, we can still distill a termination prior $p_\theta^T$ which can be used on future tasks. Instead of learning $p_\theta^T$ by minimizing the KL against the posterior termination policies, we can get more decisive terminations by minimizing

$$\min_\theta \sum_{i=1}^n \mathbb{E}_{\tau \sim q^i} \left[ \mathbb{D}_{\text{KL}} \big( \hat{q}_{\phi_i}(\cdot|s_t, z_{t-1}) \| p_\theta^T(\cdot|s_t, z_{t-1}) \big) \right], \tag{7}$$

and $\hat{q}_{\phi_i}(b=1|s_t, z_{t-1}) = \sum_{z_t \neq z_{t-1}} q_{\phi_i}^H(z_t|s_t, z_{t-1}, b_t=1)$ i.e., the learned termination prior distills the probability that the tasks' master policies would change the active option if they had the opportunity. Details on how we optimized the MSOL objective are given in Appendix C.

## 4  RELATED WORK

Most hierarchical approaches rely on proxy rewards to train the lower level components and their terminations. Some of them aim to reach pre-specified subgoals (Sutton et al., 1999; Kulkarni et al., 2016), which are often found by analyzing the structure of the MDP (McGovern & Barto, 2001; Menache et al., 2002; Şimşek et al., 2005; Şimşek & Barto, 2009), previously learned policies (Goel & Huber, 2003; Tessler et al., 2017) or predictability (Harutyunyan et al., 2019). Those methods typically require knowledge, or a sufficient approximation, of the transition model, both of which are often infeasible.

Recently, several authors have proposed unsupervised training objectives for learning diverse skills based on their distinctiveness (Gregor et al., 2016; Florensa et al., 2017; Achiam et al., 2018; Eysenbach et al., 2019). However, those approaches don't learn termination functions and cannot guarantee that the required behavior on the downstream task is included in the set of learned skills. Hausman et al. (2018) also incorporate reward information, but do not learn termination policies and are therefore restricted to learning multiple solutions to the provided task instead of learning a *decomposition* of the task solutions which can be re-composed to solve new tasks.

A third usage of proxy rewards is by training lower level policies to move towards goals defined by the higher levels. When those goals are set in the original state space (Nachum et al., 2018a), this approach has difficulty scaling to high dimensional state spaces like images. Setting the goals in a learned embedding space (Dayan & Hinton, 1993; Vezhnevets et al., 2017; Nachum et al., 2018b) can be difficult to train, though. In both cases, the temporal extension of the learned skills are set manually. On the other hand, Goyal et al. (2019) also learn a hierarchical agent, but not to transfer skills, but to find decisions states based on how much information is encoded in the latent layer.

HiREPS Daniel et al. (2012) also take an inference motivated approach to learning options. In particular Daniel et al. (2016) propose a similarly structured hierarchical policy, albeit in a single task setting. However, they do not utilize learned prior *and* posterior distributions, but instead use expectation maximization to iteratively infer a hierarchical policy to explain the current reward-weighted trajectory distribution.

Several previous works try to overcome the restrictive nature of options that can lead to sub-optimal solutions by allowing the higher-level actions to modulate the behavior of the lower-level policies Schaul et al. (2015); Heess et al. (2016); Haarnoja et al. (2018). However, this significantly increases the required complexity of the higher-level policy and therefore the learning time.

The multitask- and transfer-learning setup used in this work is inspired by Thrun & Schwartz (1995) and Pickett & Barto (2002) who suggest extracting options by using commonalities between solutions to multiple tasks. Prior multitask approaches often rely on additional human supervision like policy sketches (Andreas et al., 2017) or desirable sub-goals (Tessler et al., 2017; Konidaris & Barto, 2007; Mann et al., 2015) in order to learn skills which transfer well between tasks. In contrast, our work aims at finding good termination states without such supervision. Tirumala et al. (2019) investigate the use of different priors for the higher-level policy while we are focussing on learning transferrable option priors. Closest to our work is Meta Learning of Shared Hierarchies (MLSH) (Frans et al., 2018) which, however, shares the lower-level policies across all tasks without

distinguishing between prior and posterior and does not learn termination policies. As discussed, this leads to local minima and insufficient diversity in the learned options. Similarly to us, Fox et al. (2016) differentiate between prior and posterior policies on multiple tasks and utilize a KL-divergence between them for training. However, they do not consider termination probabilities and instead only choose one option per task. Instead of transferring option policies between tasks, Ammar et al. (2014) aim to share behavior through a latent embedding. Another interesting approach to multitask learning is (Mankowitz et al., 2016) which learns decision regions that are linear in the state instead of learning nonlinear master- and termination policies.

Our approach is closely related to DISTRAL (Teh et al., 2017) with which we share the multitask learning of prior and posterior policies. However, DISTRAL has no hierarchical structure and applies the same prior distribution over primitive actions, independent of the task. As a necessary hierarchical heuristic, the authors propose to also condition on the last primitive action taken. This works well when the last action is indicative of future behavior; however, in Section 5 we show several failure cases where a *learned* hierarchy is needed.

## 5 EXPERIMENTS

We conduct a series of experiments to show: i) when learning hierarchies in a multitask setting, MSOL successfully overcomes the local minimum of insufficient option diversity, as described in Appendix A; ii) MSOL can learn useful termination policies; iii) MSOL is equally applicable to discrete as well as continuous domains; and iv) using soft options yields fast transfer learning while still reaching optimal performance, even on new, out-of-distribution tasks.

All architectural details and hyper-parameters can be found in the appendix. For all experiments, we first train the exploration priors and options on $n$ tasks from the available task distribution $\mathcal{T}$ (training phase is plotted in Appendix E). Subsequently, we test how quickly we can learn new tasks from $\mathcal{T}$ (or another distribution $\mathcal{T}'$).

We compare the following algorithms: MSOL is our proposed method that utilizes soft options both during option learning and transfer. MSOL(frozen) uses the soft options framework during learning to find more diverse skills, but does not allow fine-tuning the posterior sub-policies after transfer. DISTRAL (Teh et al., 2017) is a strong non-hierarchical transfer learning algorithm that also utilizes prior and posterior distributions. DISTRAL(+action) utilizes the last action as option-heuristic which works well in some tasks but fails when the last action is not sufficiently informative. MLSH (Frans et al., 2018) is a multitask option learning algorithm like MSOL, but utilizes 'hard' options for both learning and transfer, i.e., sub-policies that are shared exactly across tasks. It also relies on fixed option durations and requires a complex training schedule between master and intra-option policies to stabilize training. We use the author's MLSH implementation. Lastly, we compare against Option Critic (OC) (Bacon et al., 2017), which takes the task-id as additional input in order to apply it to a multitask setting.

### 5.1 MOVING BANDITS

We start with the 2D Moving Bandits environment proposed and implemented by Frans et al. (2018), which is similar to the example in Appendix A. In each episode, the agent receives a reward of 1 for each time step it is sufficiently close to one of two randomly sampled, distinguishable, marked positions in the environment. The agent can take actions that move it in one of the four cardinal directions. Which position is not signaled in the observation. Each episode lasts 50 time steps.

We compare against MLSH and DISTRAL to highlight challenges that arise in multitask training. We allow MLSH and MSOL to learn two options. During transfer, optimal performance can only be achieved when both options successfully learned to reach *different* marked locations, i.e., when they are diverse. In Figure 2(a) we can see that MSOL is able to do so but the options learned by MLSH are not sufficiently diverse, for the reason explain in Appendix A. DISTRAL, even with the last action provided as additional input, is not able to quickly utilize the prior knowledge. The last action only conveys meaningful information when taking the goal locations into account: DISTRAL agents need to *infer* the intention based on the last action and the relative goal positions. While this is possible, in practice the agent was not able to do so, even with a much larger network. However, longer training allows DISTRAL to perform as well as MSOL, since its posterior is flexible, denoted

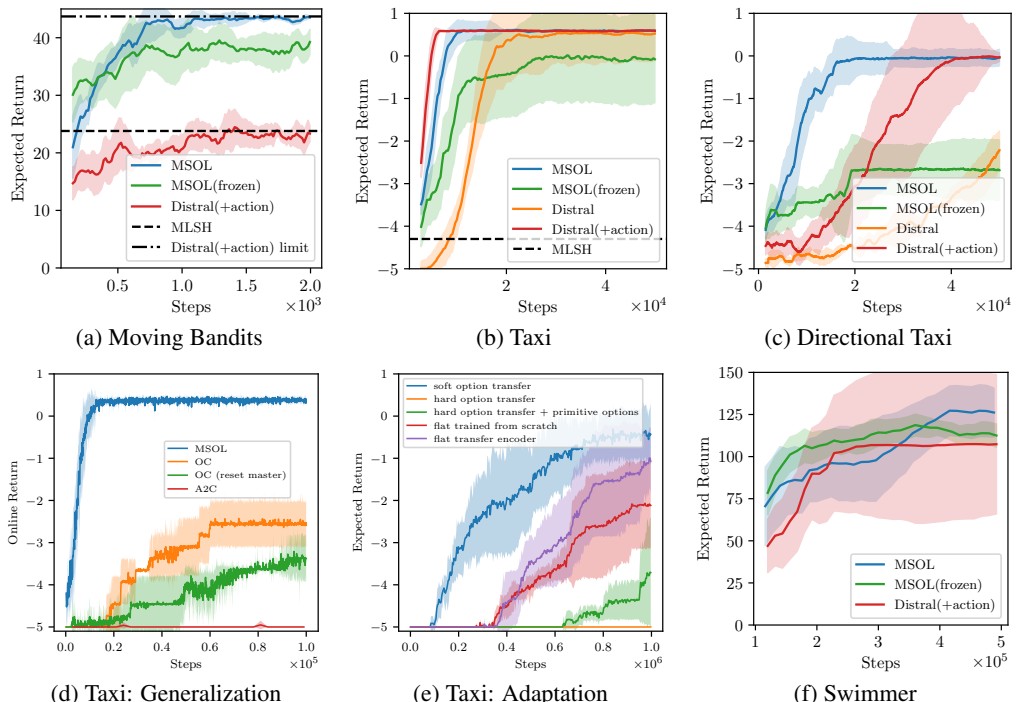

Figure 2: Performance during testing of the learned options and exploration priors. Each line is the median over 5 random seeds (2 for MLSH) and shaded areas indicate standard deviations. More details in text.

by "DISTRAL(+action) limit". Lastly, MSOL(frozen) also outperforms DISTRAL(+action) and MLSH, but performs worse that MSOL. This highlights the utility of making options soft, i.e. adaptable.

## 5.2 TAXI

Next, we use a slightly modified version of the original Taxi domain (Dietterich, 1998) to show learning of termination functions as well as transfer- and generalization capabilities. To solve the task, the agent must pick up a passenger on one of four possible locations by moving to their location and executing a special 'pickup/drop-off' action. Then, the passenger must be dropped off at one of the other three locations, again using the same action executed at the corresponding location. The domain has a discrete state space with 30 locations arranged on a grid and a flag indicating whether the passenger was already picked up. The observation is a one-hot encoding of the discrete state, excluding passenger- and goal location. This introduces an information-asymmetry between the task-specific master policy, and the shared options, allowing them to generalize well (Galashov et al., 2018). Walls (see Figure 3) limit the movement of the agent and invalid actions.

We investigate two versions of Taxi. In the original (Dietterich, 1998, just called *Taxi*), the action space consists of one no-op, one 'pickup/drop-off' action and four actions to move in all cardinal directions. In *Directional Taxi*, we extend this setup: the agent faces in one of the cardinal directions and the available movements are to move forward or rotate either clockwise or counter-clockwise.

In both environments the set of tasks $\mathcal{T}$ are the 12 different combinations of pickup/drop-off locations. Episodes last at most 50 steps and there is a reward of 2 for delivering the passenger to its goal and a penalty of -0.1 for each time step. During training, the agent is initialized to any valid state. During testing, the agent is always initialized without the passenger on board.

We allow four learnable options in MLSH and MSOL. This necessitates the options to be diverse, i.e., one option to reach each of the four pickup/drop-off locations. Importantly, it also requires the options to learn to terminate when a passenger is picked up. As one can see in Figure 2(b), MLSH struggles both with option-diversity and due to its fixed option duration which is not flexible enough for this environment. DISTRAL(+action) performs well in the original *Taxi* environment, as seen in Figure 2(b), since here the last action is a good indicator for the agent's intention. However, in the directional case shown in Figure 2(c), the actions are less informative and make it much harder

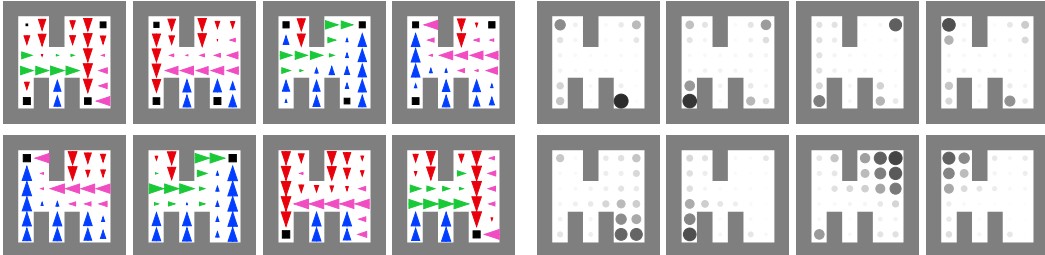

Figure 3: Options learned with MSOL on the taxi domain, before (top) and after pickup (bottom). The light gray area indicates walls. The left plots show the intra-option policies: arrows and colors indicated direction of most likely action, the size indicates its probability. A square indicates the pickup/dropoff action. The right plots show the termination policies: intensity and size of the circles indicate termination probability.

for DISTRAL to use prior knowledge. By contrast, MSOL performs well in both taxi environments. Comparing its performance with MSOL(frozen) shows the utility of adaptable soft options during transfer.

Figure 3, which visualizes the options learned by MSOL, shows that it successfully learns useful movement primitives and termination functions. The same soft option represents different behavior depending on whether it already picked up the passenger. This is expected as this behavior does not need to terminate the current option on three of the 12 tasks.

Next we show how learning information-asymmetric soft options can help with transfer to unseen tasks. In Figure 2(d) we show learning on four tasks from $\mathcal{T}$ using options that were trained on the remaining eight, comparing against A2C and OC. Note that in OC, there is no information-asymmetry: We share the same networks across all tasks and provide the task-id as additional input, including to the option-policies. This prevents them from generalizing well to unseen tasks. On the other hand, withholding the task-information from them would be similar to MLSH, which we already showed to struggle with local minima. The strong performance of MSOL on this task shows that we need *soft* options to be able to train information-asymmetric options that generalize well.

We also investigate the utility of flexible soft options: In Figure 2(e) we show learning performance on twelve *changed* tasks in which the pickup/dropoff locations where moved by one cell while the options were trained with the original locations. As expected, hard options are not able to solve this tasks. Even with additional access to primitive actions, exploration is inefficient (Jong et al., 2008). On the other hand, MSOL is able to quickly learn this new task by adapting the previously learned options, outperforming hard options and flat policies learned from scratch.

## 5.3 SWIMMER

Lastly, we show that MSOL can also be applied to continuous multitask domains. In particular, we investigate the MuJoCo environment 'Swimmer' (Todorov et al., 2012; Brockman et al., 2016). Instead of rewarding forward movement as in the original implementation, now the rewarded movement direction depends on the task from $\mathcal{T} = \{up, down, left, right\}$. We also include a small amount of additive action noise (details in the Appendix). We show that MSOL performs competitive even in the absence of known failure cases of DISTRAL (see Figure 2(f)).

## 6 DISCUSSION

Multitask Soft Option Learning (MSOL) proposes reformulating options using the perspective of prior and posterior distributions. This offers several key advantages.

First, during transfer, it allows us to distinguish between fixed, and therefore knowledge-preserving option *priors*, and flexible option *posteriors* that can adjust to the reward structure of the task at hand. This effects a similar speed-up in learning as the original options framework, while avoiding sub-optimal performance when the available options are not perfectly aligned to the task. Second, utilizing this 'soft' version of options in a multitask learning setup increases optimization stability and removes the need for complex training schedules between master and lower level policies. Furthermore, this framework naturally allows master policies to coordinate across tasks and avoid local minima of insufficient option diversity. It also allows for autonomously learning option-termination policies, a very challenging task which is often avoided by fixing option durations manually.

Lastly, using this formulation also allows inclusion of prior information in a principled manner without imposing too rigid a structure on the resulting hierarchy. We utilize this advantage to explicitly incorporate the bias that good options should be temporally extended. In future research, other types of information can be explored. As an example, one could investigate sets of tasks which would benefit from a learned master prior, like walking on different types of terrain.

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

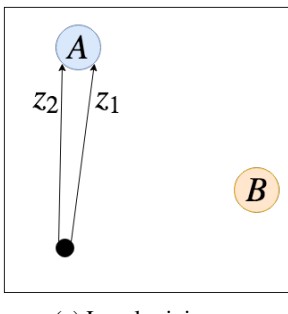 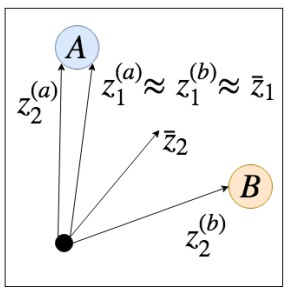 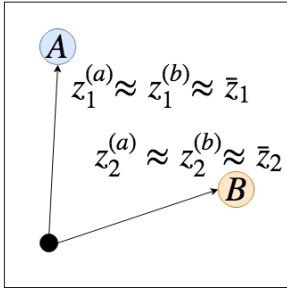

(a) Local minimum        (b) Using skill priors        (c) After training

Figure 4: Hierarchical learning of two concurrent tasks ($a$ and $b$) using two options ($z_1$ and $z_2$) to reach two relevant targets ($A$ and $B$). a) Local minimum when simply sharing options across tasks. b) Escaping the local minimum by using prior ($\bar{z}_i$) and posterior ($z_i^{(j)}$) policies. c) Learned options after training. Details are given in the text in Appendix A.

# Appendix

## A    LOCAL MINIMA IN MULTITASK OPTION LEARNING

Here our aim is to provide an intuitive explanation of why learning hard options in a multitask setting can lead to local minima. In this local minimum, multiple options have learned the same behavior and are unable to change it, even if doing so would ultimately lead to a higher reward. We use the Moving Bandits experiment schematically depicted in Figure 4 as example: The agent (black dot) observes two target locations $A$ and $B$ but does not know which one is the correct one that has to be reached in order to generate a reward.

The state- and action-spaces are continuous, requiring multiple actions to reach either $A$ or $B$ from the starting position. Consequently, having access to two options, one for each location, can accelerate learning. We denote the options by $z_1$ and $z_2$.

Assume that due to random initialization or later discovery of target $B$, both skills currently reach $A$. Furthermore, assume that we are training on multiple task *simultaneously* as often the case in RL. In this situation, the master policies on tasks in which the correct goal is $A$ are indifferent between using $z_1$ and $z_2$ and will consequently use *both* with high probability.

Now, changing one skill, e.g. $z_2$, towards $B$ in order to solve tasks in which $B$ is the correct target, decreases the performance on all tasks that currently use $z_2$ to reach target $A$ because for hard options the skills are shared exactly across tasks. Due to small, stochastic gradient updates changing $z_2$ towards $B$ only slightly increases the probability of reaching $B$ but significantly decreases the probability of reaching $A$ (because now the option $z_2$ will at first point to neither $A$ or $B$). Averaged across all tasks, this would at first *decrease* the average reward, consequently preventing any option from changing away from $A$, leaving $B$ unreachable.

To "free up" $z_2$ (or any other option) and learn a new skill, all master policies would need to coordinate and refrain from using it to reach $A$ and instead use the equally useful skill $z_1$ exclusively. Importantly, using soft options makes this possible.

In Figures 4(b) and 4(c) we depict this schematically: The key difference is that in MSOL we have separate *task-specific posteriors* $z_i^{(a)}$ and $z_i^{(b)}$ for tasks $a$ and $b$ and soft options $i \in \{1, 2\}$ (for simplicity, we assume that the correct target is $A$ for task $a$ and $B$ for task $b$). This allows us, in a first step, to solve *all* tasks (Figure 4(b)). Later, as discussed in Section 3.3, maximizing the regularized reward (6) will lead to specialized skills, i.e. option *priors* (Figure 4(c)).

The corresponding shared priors $\bar{z}_i$ (one per option) are updated to effectively yield the 'average' (as measured by the KL-divergence) between the task-specific posteriors $z_i^{(a)}$ and $z_i^{(b)}$. Allowing the posteriors of an option, for example $z_2^{(a)}$ and $z_2^{(b)}$, to deviate from the prior $\bar{z}_2$ allows that option, in a first step, to solve *both* tasks, irrespective of the misaligned priors. This is shown in Figure 4(b)

where all posteriors of $z_1$ are moving to $A$, but for $z_2$, the posteriors on different tasks $a$ and $b$ move to different targets.

Crucially, to maximize the regularized reward (6), the KL-divergences between priors and posteriors should be minimized along they trajectories, i.e. *weighted by how likely they are to occur*. Consequently, the higher $\mathbb{D}_{\text{KL}}[z_2^{(a)} \| \bar{z}_2] > 0$ can be avoided on task $a$ by picking the more specialized option $z_1$ for which $\mathbb{D}_{\text{KL}}[z_1^{(a)} \| \bar{z}_1] = 0$. This, in turn, "frees up" $z_2$ to now also specialize to $B$ as shown in Figure 4(c). After this implicit coordination, $z_1^{(a)}$ is always used to reach $A$ and $z_2(b)$ is always used to reach $B$, allowing both $\bar{z}_1$ and $\bar{z}_2$ to also lead to $A$ and $B$ respectively.

## B  RELATIONSHIP BETWEEN MSOL AND CLASSICAL OPTIONS

Assume we are faced with a new task and are given some prior knowledge in the form of a set of skills that we can use. Using those skills and their termination probabilities as prior policies $p^T$ and $p^L$ in the soft option framework, we can see $\beta$ as a temperature parameter determining how closely we are restricted to following them. For $\beta \to \infty$ we recover the classical 'hard' option case and our posterior option policies are restricted to the prior.[1] For $\beta = 0$ the priors only initialize the otherwise unconstrained policy, quickly unlearning behavior that may be useful down the line. Lastly, for $0 < \beta < \infty$ we use the prior information to guide exploration but are only softly restricted to the given skills and can also explore and use policies 'close' to them.

## C  MSOL TRAINING DETAILS

### C.1  OPTIMIZATION

Even though $R_i^{\text{reg}}$ depends on $\phi_i$, its gradient w.r.t. $\phi_i$ vanishes.[2] Consequently, we can treat the regularized reward as a classical RL reward and use any RL algorithm to find the optimal hierarchical policy parameters $\phi_i$. In the following, we explain how to adapt A2C (Mnih et al., 2016) to soft options. The extension to PPO (Schulman et al., 2017) is straightforward.[3]

The joint posterior policy in (4) depends on the current state $s_t$ and the previously selected option $z_{t-1}$. The expected sum of regularized future rewards of task $i$, the value function $V_i$, must therefore also condition on this pair:

$$V_i(s_t, z_{t-1}) \ := \ \mathbb{E}_{\tau \sim q} \Big[ \sum_{t'=t}^{T} \gamma^{t'-t} R_{i,t'}^{\text{reg}} \, \Big| \, s_t, z_{t-1} \Big]. \tag{8}$$

As $V_i(s_t, z_{t-1})$ cannot be directly observed, we approximate it with a parametrized model $V_{\phi_i}(s_t, z_{t-1})$. The $k$-step advantage estimation at time $t$ of trajectory $\tau$ is given by

$$A_{\phi_i}(\tau_{t:(t+k)}) \ := \ \sum_{j=0}^{k-1} \gamma^j R_{t+j}^{\text{reg}} + \gamma^k V_{\phi_i}^-(s_{t+k}, z_{t+k-1}) - V_{\phi_i}(s_t, z_{t-1}) \,, \tag{9}$$

where the superscript '$-$' indicates treating the term as a constant. The approximate value function $V_{\phi_i}$ can be optimized towards its bootstrapped $k$-step target by minimizing $\mathcal{L}_V(\phi_i, \tau_{1:T}) := \sum_{t=1}^{T} (A_{\phi_i}(\tau_{t:(t+k)}))^2$. As per A2C, $k \in [1 \dots n_s]$ depending on the state (Mnih et al., 2016). The corresponding policy gradient loss is

$$\mathcal{L}_A(\phi_i, \tau_{1:T}) := \sum_{t=1}^{T} A_{\phi_i}^-(\tau_{t:(t+k)}) \ln q_{\phi_i}(a_t, z_t, b_t | s_t, z_{t-1}) \,.$$

The gradient w.r.t. the prior parameters $\theta$ is[4]

$$\nabla_\theta \mathcal{L}_P(\theta, \tau_{1:T}, \tilde{b}_{1:T}) \ := \ -\sum_{t=1}^{T} \Big( \nabla_\theta \ln p_\theta^L(a_t | s_t, z_t) + \nabla_\theta \ln p_\theta^T(\tilde{b}_t | s_t, z_{t-1}) \Big), \tag{10}$$

---

[1] However, in this limiting case optimization using the regularized reward is not possible.

[2] $\int p(x) \nabla \ln p(x) \, dx = \int \nabla p(x) \, dx = \nabla \int p(x) \, dx = 0$.

[3] However, for PAI frameworks like ours, unlike in the original PPO implementation, the advantage function must be updated after each epoch.

[4] Here we ignore $\beta$ as it is folded into $\lambda_P$ later.

where $\tilde{b}_t = \delta_{z_{t-1}}(z'_t)$ and $z'_t \sim q^H(z'_t|s_t, z_{t-1}, b_t = 1)$. To encourage exploration in all policies of the hierarchy, we also include an entropy maximization loss:

$$\mathcal{L}_H(\phi_i, \tau_{1:T}) \quad := \quad \sum_{t=1}^{T} \Big( \ln q_{\phi_i}^H(z_t|s_t, z_{t-1}, b_t) + \ln q_{\phi_i}^L(a_t|s_t, z_t) + \ln q_{\phi_i}^T(b_t|s_t, z_{t-1}) \Big). \quad (11)$$

Note that term ① in (6) already encourages maximizing $\mathcal{L}_H(\phi_i, \tau)$ for the master policy, since we chose a uniform prior $p^H(z_t|b_t = 1)$. As both terms serve the same purpose, we are free to drop either one of them. In our experiments, we chose to drop the term for $q^H$ in $R_t^{\text{reg}}$, which proved slightly more stable to optimize that the alternative.

We can optimize all parameters jointly with a combined loss over all tasks $i$, based on sampled trajectories $\tau^i := \tau_{1:T}^i \sim q_{\phi_i}$ and corresponding sampled values of $\tilde{b}^i := \tilde{b}_{1:T}^i$:

$$\mathcal{L}(\{\phi_i\}, \theta, \{\tau^i\}, \{\tilde{b}^i\}) = \sum_{i=1}^{n} \Big( \mathcal{L}_A(\phi_i, \tau^i) + \lambda_V \mathcal{L}_V(\phi_i, \tau^i) + \lambda_P \mathcal{L}_P(\theta, \tau^i, \tilde{b}^i) + \lambda_H \mathcal{L}_H(\phi_i, \tau^i) \Big).$$

### C.2 TRAINING SCHEDULE

For faster training, it is important to prevent the master policies $q^H$ from converging too quickly to allow sufficient updating of all options. On the other hand, a lower exploration rate leads to more clearly defined options. We consequently anneal the exploration bonus $\lambda_H$ with a linear schedule during training.

Similarly, a high value of $\beta$ leads to better options but can prevent finding the extrinsic reward $r_i(s_t, a_t)$ early on in training. Consequently, we increase $\beta$ over the course of training, also using a linear schedule.

## D ARCHITECTURE

All policies and value functions share the same encoder network with two fully connected hidden layers of size 64 for the Moving Bandits environment and three hidden layers of sizes 512, 256, and 512 for the Taxi environments. Distral was tested with both model sizes on the Moving Bandits task to make sure that limited capacity is not the problem. Both models resulted in similar performance, the results shown in the paper are for the larger model. On swimmer the encoder model size is $1024 \times 256 \times 512$. Master-policies, as well as all prior- and posterior policies and value functions consist of only one layer which takes the latent embedding produced by the encoder as input. Furthermore, the encoder is shared across tasks, allowing for much faster training since observations can be batched together.

Options are specified as an additional one-hot encoded input to the corresponding network that is passed through a single 128 dimensional fully connected layer and concatenated to the state embedding before the last hidden layer. We implement the single-column architecture of Distral as a hierarchical policy with just one option and with a modified loss function that does not include terms for the master and termination policies. Our implementation builds on the A2C/PPO implementation by Kostrikov (2018), and we use the implementation for MLSH that is provided by the authors (https://github.com/openai/mlsh).

## E HYPER-PARAMETERS AND ADDITIONAL ENVIRONMENT DETAILS

We use $2\lambda_V = \lambda_A = \lambda_P = 1$ in all experiments. Furthermore, we train on all tasks from the task distribution, regularly resetting individual tasks by resetting the corresponding master and re-initializing the posterior policies. Optimizing $\beta$ for MSOL and Distral was done over $\{0.01, 0.02, 0.04, 0.1, 0.2, 0.4\}$. We use $\gamma = 0.95$ for Moving Bandits and Taxi and $\gamma = 0.995$ for Swimmer.

### E.1 MOVING BANDITS

For MLSH, we use the original hyper-parameters (Frans et al., 2018). The duration of each option is fixed to 10. The required warm-up duration is set to 9 and the training duration set to 1. We also

use 30 parallel environments split between 10 tasks. This and the training duration are the main differences to the original paper. Originally, MLSH was trained on 120 parallel environments which we were unable to do due to hardware constraints. Training is done over 6 million frames per task.

For MSOL and Distral we use the same number of 10 tasks and 30 processes. The duration of options are learned and we do not require a warm-up period. We set the learning rate to $0.01$ and $\beta = 0.2$, $\alpha = 0.95$, $\lambda_H = 0.05$. Training is done over 0.6 million frames per task. For Distral we use $\beta = 0.04$, $\lambda_H = 0.05$ and also 0.6 million frames per task.

## E.2 Taxi

For MSOL we anneal $\beta$ from 0.02 to 0.1 and $\lambda_H$ from 0.1 to 0.05. For Distral we use $\beta = 0.04$. We use 3 processes per task to collect experience for a batch size of 15 per task. Training is done over 1.4 million frames per task for *Taxi* and 4 million frames per task for *Directional Taxi*. MLSH was trained on 0.6 million frames for *Taxi* as due to it's long runtime of several days, using more frames was infeasible. Training was already converged.

## E.3 Taxi: Adaptation to changed tasks

As shown in the main experiments, soft options can still be useful, even when the task distribution changes. It is unsurprising that hard options are not able to solve the task. However, interestingly, providing hard options *and* primitive actions to the master policy performs much worse than just learning from scratch. This phenomenon was investigated previously by Jong et al. (2008) and further justifies using soft options for transfer to out-of-distribution tasks.

Whether training from scratch or re-using misspecified options that were trained on a different set of tasks is learning faster mainly depends on i) how strongly the options are misspecified and ii) how difficult the exploration problem is in the environment. This tradeoff is shown in Figure 6: On a smaller 8x8 grid (left), learning from scratch performs competitively because exploration is sufficiently simple. On the other hand, on a 10x10 grid (right, results from the main paper), exploration is harder and soft options allow for significantly faster learning because they can guide the exploration in a helpful way.

## E.4 Swimmer

For training Distral and MSOL we use PPO instead of A2C as it generally achieves better performance on continuous tasks. We have $\lambda_H = 0.0004$ for both MSOL and Distral for primitive actions and $\lambda'_H = 0.1$ for the master- and termination policies in MSOL. We use a learning rate of 0.0002, GAE (Schulman et al., 2015) with $\tau = 0.98$. We collect 2000 steps in parallel on 6 processes per task, resulting in a batchsize of $12,000$ per task. Training is done over 6 million frames with a linearly scheduled increase of $\beta$ from 0 to 0.04 for MSOL and 0.01 for Distral. We set $\alpha = 0.98$.

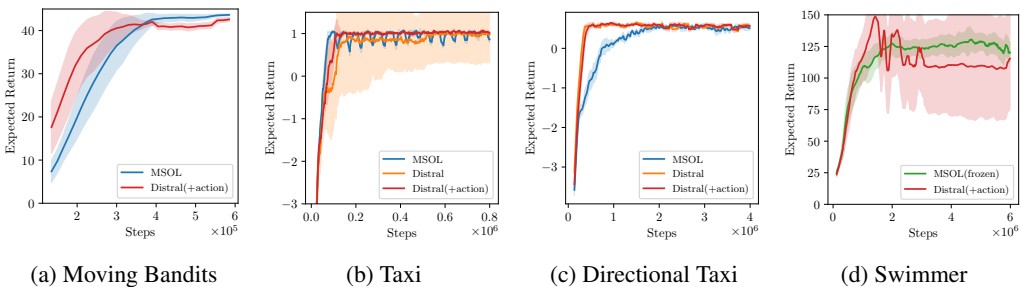

(a) Moving Bandits     (b) Taxi     (c) Directional Taxi     (d) Swimmer

Figure 5: Performance during training phase. Note that MSOL and MSOL(frozen) share the same training as they only differ during testing. Further, note that the highest achievable performance for Taxi and Directional Taxi is higher during training as they can be initialized closer to the final goal (i.e. with the passenger on board).

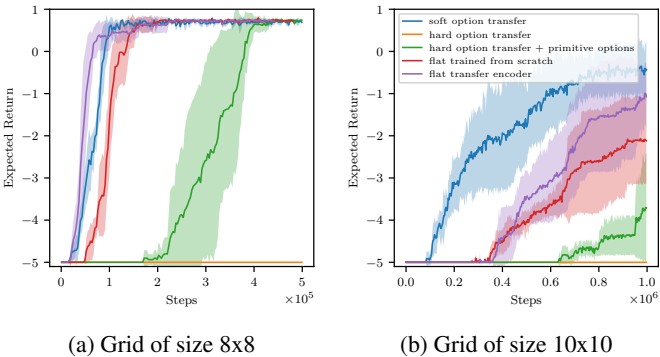

(a) Grid of size 8x8        (b) Grid of size 10x10

Figure 6: Tradeoff between exploration and misspecification: For smaller grid sizes, exploration is easier and learning from scratch performs competitively. For larger grid sizes, soft options can accelerate training through faster exploration, even if they are misspecified because they were trained on a different set of tasks.

