# OpenReview forum: "Multitask Soft Option Learning"
_ICLR.cc/2020/Conference — Reject_

### Official Review · AnonReviewer3 · 2019-10-16
**Official Blind Review #3**

**Rating:** 8

**Review:**

This paper is about learning hierarchical multitask policies over options.  The hierarchical prior policy is shared amongst all tasks, while the hierarchical posterior policies are allowed to adapt to specific tasks.  Once the prior is learned, it is fixed.   The parameters of the posterior policies are adjusted via an adapted version of A2C.

I liked the flow and the organization of this paper.

Comments and questions:
* I see that term 1 and term 3 of equation (6) working together to ensure some kind of exploration and exploitation.  Term 2 controls how the option posterior deviates from the prior.  However, when the ratio is 1 or less than 1, the value of (6) would increase, and both cases would have made the posterior more like the prior.  There seems to be no other term that incentivizes the option posterior to deviate, and I do not see how the options are adapting to tasks.

* The term 1,2,3 in (6) are weighted equally by beta and cannot be fine-tuned to desired trade-offs.

* Should there be \xi(i) multiplying the last term in (3)?

* What does delta(z_t - z_{t-1}) mean in section 3.1?  It was not defined anywhere.

Misspelling and typos:
* page 5: optimized is misspelled in "Details on how we optimiszed..."

* Appendix C.1: should there not be a superscript dash on A_{\pi_i} since the superscript dash carries the meaning that the term is a constant.

**Experience Assessment:**

I have read many papers in this area.

**Review Assessment: Checking Correctness Of Derivations And Theory:**

I assessed the sensibility of the derivations and theory.

**Review Assessment: Checking Correctness Of Experiments:**

I assessed the sensibility of the experiments.

**Review Assessment: Thoroughness In Paper Reading:**

I read the paper thoroughly.

---

> ### Author Response · Authors · 2019-11-12
> **Response to Review #3**
>
> Thank you for the detailed reading of our work and your feedback. We hope we are able to address your comments below and welcome additional questions or comments.
>
> Equation 6:
> The incentive to deviate from the option-prior comes from the first term in equation 6 (which we didn’t assign a number to), the reward $r_i$.
> So overall, the posterior tries to optimize the reward (which might try to pull the posterior away from the prior) while deviating from the three priors as little as possible, which induces the various effects for terms 1,2,3.
>
> Equal value of $\beta$ for terms 1,2,3:
> We chose to restrict ourselves to the same value of $\beta$ for all three terms to show that our algorithm works well even in this restricted case.
> However, you are right that this is restrictive and it is quite likely that it is possible to further improve the results of MSOL by fine-tuning all three values separately. However, this would make the comparison to Distral less fair as it only has one regularization hyperparameter.
> Consequently, we believe that showing superior performance for equal values of $\beta$ is a stronger argument in favor of MSOL.
>
> Xi in Term 3: Yes, thank you!
>
> Delta(z_t-z_{t-1}):
> Thank you for pointing this out; we included a definition in the text.
> Since we assume $z_t$ to be discrete, the delta here has the meaning of a Kronecker delta.
> For continuous $z_t$, it would correspond to the Dirac Delta distribution.
> In both cases we mean to express that $z_t=z_{t-1}$ with probability 1. In other words, if $b_t=0$, i.e. when the option doesn’t terminate, we don’t change the active option.
>
> Typo:
> Thank you for pointing this out.
>
> Appendix C.1:
> We believe you are referring to equation 9?
> We chose not to include the superscript dash here because we are using $A_{\phi_i}$ twice, once in $\mathcal{L}_A$ in which case it is indeed assumed constant. We are also using it in $\mathcal{L}_V$, in which case it is not constant as it is optimized w.r.t $V_{\phi_i}(s_t, z_{t-1})$, i.e. the last term in equation 9.

---

### Official Review · AnonReviewer2 · 2019-10-23
**Official Blind Review #2**

**Rating:** 3

**Review:**

Summary:
The authors propose a method for learning hierarchical policies in a multi-task setting built on options and casts the concepts into the Planning as Inference framework. They claim that the method brings several advantages such as:
-Stabilize end-to-end mtl,
-Leads to coordination b/w master policies,
-Allows fine-tuning of options
-Learn termination policies naturally

The proposed approach derives the reward Eq.6 by extending the graph to options for Levine’s tutorial. Eq.6 is simply the extension of the reward of maximum entropy RL to the options framework. The ideas presented in the paper are interesting, but I have concerns about the scalability of such an approach. Please see the detailed comments below.  Additionally, please note that although I have marked a weak reject, I am open to adjusting my score if the rebuttal addresses enough issues.

Detailed Comments:
A primary weakness of this approach is that it seems like there is one network that learns the options and is shared across all task (that would be the prior) and then there is a task-specific network for all options (posterior), wouldn’t this be very difficult to scale if we want to learn reusable options over the lifetime of an agent? If there are n tasks, do you need to use n different networks?

The authors assume that all options are present everywhere i.e. I ⊆ S. I think the work could benefit from removing this assumption.

The authors mention that unlike (Frans et al., 2018), they learn both intra-option and termination policies: there is definitely more work that aims to learn both the skill and its termination. It would be more complete to cite additional references here that learn both of these or rephrase this sentence.

It does not seem clear why “term 1 of the regularization is only nonzero whenever we terminate an option and the master policy needs to select another to execute.” This doesn’t seem true as this is a ratio of the two probabilities and not just the instantiation of the random variable.

The results in moving bandits alone are very convincing. However, in Taxi (2b) distral+action seems to be as good as/even better MSOL. In directional Taxi (2c) Distral(+action) manages to reach the same final performance (if we care about that), can you please comment on this.

Some parts of the experiments section does not seem clear to me, Does the proposed approach use a network per task? if yes, then it is obvious that their method could improve over learning on 12 tasks with one set of network. Please clarify.

One major concern is that the only high dimensional experiment is a swimmer and it is not immediately clear how much do we gain there. Distral is relatively closer in performance to both MSOL and MSOL frozen. I would recommend evaluation in a variety of high-dimensional domains such as other instances in Mujoco, and visual domains. In particular, the proposed ideas would make a stronger case if the baselines included other multitask hierarchical agents such as [4] for example. A discussion including some of the missing relevant related multi-task literature would also be helpful [1,2,4,5,6].

[1] Mann, T. A., Mannor, S., & Precup, D. (2015). Approximate value iteration with temporally extended actions. JAIR, 53, 375-438.
[2] Konidaris, G., & Barto, A. G. (2007). Building Portable Options: Skill Transfer in Reinforcement Learning. In IJCAI,
[3] Andreas, J., Klein, D., & Levine, S. (2017). Modular multitask reinforcement learning with policy sketches. ICML
[4] Tessler, Chen, et al. "A deep hierarchical approach to lifelong learning in minecraft." Thirty-First AAAI Conference on Artificial Intelligence. 2017.
[5] Ammar, Haitham Bou, et al. "Online multi-task learning for policy gradient methods." International Conference on Machine Learning. 2014.
[6] Mankowitz, Daniel J., Timothy A. Mann, and Shie Mannor. "Adaptive skills adaptive partitions (ASAP)." Advances in Neural Information Processing Systems. 2016.

**Experience Assessment:**

I have published one or two papers in this area.

**Review Assessment: Checking Correctness Of Derivations And Theory:**

I assessed the sensibility of the derivations and theory.

**Review Assessment: Checking Correctness Of Experiments:**

I carefully checked the experiments.

**Review Assessment: Thoroughness In Paper Reading:**

I made a quick assessment of this paper.

---

> ### Author Response · Authors · 2019-11-12
> **Response to Review #2**
>
> Thank you for your detailed review and feedback. We welcome any further comments and questions.
>
> Scalability:
> We choose to use one network per task, but this is not strictly necessary if one wanted to scale to more tasks. For example, in the non-hierarchical case with prior/posterior networks, Galashov et al. use just one posterior network, providing task information as additional input. Their findings could be applied to our hierarchical approach.
> Furthermore, we show that the derived loss function incentivises specialized options during training, i.e. options for which the prior and posterior are the same. Consequently, given sufficient options, eventually one can forget the posteriors as the option priors should be good enough to solve the tasks encountered so far.
>
> Fairness of comparison:
> Both MSOL and Distral use one network per task so this comparison is fair.
> We also included MSOL(frozen) which, once the prior options are trained, only uses this one network without further adaptation for transfer, allowing a fair comparison to MLSH, which it still outperforms.
>
> Environmental complexity and additional baselines:
> One key goal of MSOL is to find good (i.e. transferable) options and terminations without any additional human-designed prior knowledge regarding subgoals. This is challenging because during option-learning the algorithm not only needs to solve each task, but needs to find a way to segment tasks into reusable sub-tasks - all without additional human information. This is a main difference to many previous algorithms which receive additional information, for example in the form of landmarks, policy sketches or designated sub-tasks for each option.
> Our goal with the Swimmer environment was to show that our algorithm is applicable to continuous tasks. Applying the algorithm in practice to much more complex multi-task domains in MuJoCo was infeasible given our limited computational resources. Hierarchical approaches applied to such settings often receive additional prior information (like sub-tasks or goals per option as in Tessler et. al) or are not constrained to learn transferable options (like in Option Critic).
>
> Option initiation sets:
> We agree that learning to restrict the initiation sets of options is an interesting avenue for future research, but believe that learning good options and termination functions, robustly, just from a set of tasks is already a significant enough contribution.
>
> Intra-option and termination learning:
> We agree and have rephrased this sentence. We have also extended the related work section. The main difference to our work is that those approaches do not rely on a multitask setting to find good termination functions, but on other ideas, like landmarks, bottleneck states or predictability.
>
> Term 1 being nonzero:
> When b_t=0, i.e. when we do not terminate, both the prior p^H and the posterior q^H are, by definition, delta(z_t-z_{t-1}), i.e. both assign a probability of 1 to the last active option. Consequently, in this case the fraction becomes one and we have for the term: beta * ln 1 = 0
> On the other hand, if b_t=1, then the prior is uniform and the posterior is the learned posterior, leading in general to a non-zero term.
>
> Comparison to Distral:
> Final performance: We would like to note that because Distal, similarly to MSOL, learns a separate posterior policy which is regularized against the prior, it will always ultimately achieve optimal performance for weak enough regularization. So it is to be expected that both MSOL and Distral achieve the same final performance.
> Our experiments show two things:
> For learning a hierarchy, we have a more robust optimization algorithm than MLSH
> Learning a hierarchy (compared to a flat prior as in Distral or compared to a heuristic hierarchy as in Distral+Action) is useful because it can accelerate learning.
>
> We intentionally included the non-directional Taxi domain to show in which cases a simpler architecture (here Distral+Action) is sufficient for optimal transfer, so its strong performance is expected.
> The main difference of this domain is that here, passing the last action is sufficiently informative to predict the likely future behavior. It is like walking in a corridor with only one door at either end: Knowing which direction we are walking in carries the same amount of information as knowing which door we want to walk towards.
> However, in directional Taxi, knowing the last action is not as informative (because it might involve a rotation) and in Moving Bandits, one could infer the intended goal from the last action if one takes the goal positions into account. However, we found that Distral+Action was unable to learn this more complex relationship. In those cases a learned hierarchy in which options carry learned semantics outperforms Distral+Action in terms of transfer speed.
>
> Related literature:
> Thank you for the pointers to additional literature. We have included them in the related work section.

---

> > ### Comment · AnonReviewer2 · 2019-11-14
> > **Follow up comments**
> >
> > Thank you for the clarifications.
> >
> > >Fairness of comparison:
> > Thank you, that is useful.
> >
> > >Our goal with the Swimmer environment was to show that our algorithm is applicable to continuous tasks. Applying the algorithm in practice to much more complex multi-task domains in MuJoCo was infeasible given our limited computational resources.
> > Considering swimmer is the only high dimensional experiment, and that the performance in swimmer is not very convincing; I find it hard to convince myself that the approach fairs well in high dimensional domains. It would add a lot of value to this work to validate the proposed approach in other Mujoco tasks, 3D navigation environments. As presented, it does  but only in Taxi domain and marginally in Swimmer.
> >
> > >learning good options and termination functions, robustly, just from a set of tasks is already a significant enough contribution.
> > I agree that the paper presents a useful contribution. However the experiments are limited and therefore other high-dimensional environments could add value and strengthen the contributions further.

---

> > > ### Author Response · Authors · 2019-11-15
> > > **Thank you for your prompt reply**
> > >
> > > Thank you for your prompt reply and for valuing the contribution of our work.
> > >
> > > We would first like to note that on Swimmer MSOL performs comparably with Distral, which is itself a strong multitask baseline.
> > > Furthermore, we argue that hierarchical methods such as ours are best suited to domains that exhibit a strong compositional structure, like the Taxi domain does. And while Swimmer is indeed a higher dimensional domain, it does not exhibit such compositional structure. As a consequence, comparable performance against Distral is a positive mark of MSOL's general utility with higher-dimensional tasks, with the expectation that where compositional structure may be leveraged, it can do so (cf. Taxi experiments).
> > >
> > > Nonetheless, while we agree in principle with your observation that additional high-dimensional experiments would further strengthen our claims, we observe that practically, the amount of compute required to be able to do is beyond our abilities. Note that it is not simply a case of being able to run a new environment, but to be able to control for all the confounds in the form of hyperparameters and general variance in RL experiments, to be able to properly assign credit/blame to our proposed changes. Particularly, compositional tasks when combined with continuous control, tend to become extremely complex, requiring a lot of resources to train---as in Ant-Gather, for example.
> > >
> > > Altogether, we believe that the experiments we perform provide good evidence for the suitability and value of MSOL for multitask learning, and hope that the lack of more complex experiments, which were not possible due to resource constraints, do not adversely affect the value of our work.

---

> > > > ### Comment · AnonReviewer2 · 2019-11-15
> > > > **Question**
> > > >
> > > > Thank you for your comments.
> > > >
> > > > >Nonetheless, while we agree in principle with your observation that additional high-dimensional experiments would further strengthen our claims, we observe that practically, the amount of compute required to be able to do is beyond our abilities.
> > > >
> > > > Would it be possible to comment on how much time does one run of the algorithm takes in an environment like Ant-Gather for example? I am curious to understand how hard is it to train with the proposed approach.

---

### Decision · Program_Chairs · 2019-12-19

**Decision:**

Reject

**Comment:**

Apologies for only receiving two reviews. R2 gave a WR and R3 gave an A. Given the lack of 3rd review and split nature of the scores, the AC has closely scrutinized the paper/reviews/comments/rebuttal. Thoughts:
 - Paper is on interesting topic.
 - AC agrees with R2's concern about the evaluation not using more complex environments like Mujoco. Without evaluation on a standard benchmark, it is difficult to know objectively if the approach works.
 - AC agrees with authors that the DISTRAL approach forms a strong baseline.
 - Nevertheless, the experiments aren't super compelling either.
 - AC has some concerns about scaling issues w.r.t. model size & #tasks.

The paper is very borderline, but the AC sides with R2's concerns and unfortunately feels the paper cannot be accepted without a stronger evaluation. With this, it would make a compelling paper.